# Natural Diatomite Supported Zirconium-Doped TiO_2_ with Tailoring Band Structure for Enhanced Visible-Light Photocatalytic Properties

**DOI:** 10.3390/nano12162827

**Published:** 2022-08-17

**Authors:** Fang Yuan, Chunquan Li, Xiangwei Zhang, Renfeng Yang, Zhiming Sun

**Affiliations:** School of Chemical and Environmental Engineering, China University of Mining and Technology (Beijing), Beijing 100083, China

**Keywords:** Zr-doped, diatomite, TiO_2_, visible light photocatalysis, tetracycline

## Abstract

The development of economically applicable, highly efficient and low cost photocatalytic materials has always been a challenge. In this work, we report a zirconium doped TiO${}_{2}$/diatomite (ZrTD) composite with enhanced visible light-induced photocatalytic activity. The as-prepared samples were characterized by X-ray diffraction, scanning electron microscopy, UV–VIS diffused reflectance spectroscopy, high-performance liquid chromatography-mass spectrometry, photoluminescence and X-ray photoelectron spectroscopy, respectively. The optimal doping ratio of zirconium into TiO${}_{2}$ was obtained at 3% (3%ZrTD composite), and the degradation rate constant of which tetracycline (TC) is up to around 8.65 times higher that of zirconium doped TiO${}_{2}$. In addition, zirconium doping introduces the impurity levels of Zr 3d and oxygen vacancies into the lattice of TiO${}_{2}$, resulting in broadening the light absorption range, reducing the band gap, and improving the separation efficiency of photogenerated electron-hole pairs, thus endowing with visible light photocatalytic properties. Moreover, both the photogenerated holes (h${}^{+}$) and superoxide (•O${}_{2}$${}^{-}$) radicals are responsible for the degradation process of TC, and a possible degradation pathway and the corresponding intermediate products of TC by ZrTD composite are also proposed in detail.

## 1. Introduction

Antibiotics, as antibacterial and growth promoting drugs, can effectively protect human health and promote the development of animal husbandry [1]. However, due to the extensive use of antibiotics, how to properly dispose of the wastewater containing antibiotics from production has become a global environmental pollution problem [2,3]. Tetracycline (TC), a common spectrum antibiotic, is one of the most widely used antibiotics at present whose annual output can reach thousands of tons [4,5]. The biological toxicity test results of TC show that they are highly or moderately toxic chemicals to aquatic algae, and it also inevitably leads to drug resistance to human pathogenic bacteria and stimulation to human organs, thus endangering human health [6,7]. Tetracycline antibiotics have been listed in the list of emerging pollutants by the US Environmental Protection Agency (USEPA) [8]. Therefore, revealing the mechanism of high-efficiency removal and degradation of tetracycline in water has been regarded as a key scientific problem to be solved urgently.

At present, the removal methods of antibiotics mainly include physical methods (adsorption [9,10], membrane filtration [11], etc.), chemical methods (ozonation [12], electrochemical oxidation [13], photocatalysis [14,15,16], etc.), biological methods (biosurfactant treatment [17], activated sludge degradation [18,19], etc.). However, TC can inhibit biological activity, making the compound difficult to remove by traditional activated sludge processes [18,19]. Meanwhile, most physicochemical methods are also limited by treatment effect, operation cost and other issues [11,19]. Hence, photocatalytic oxidation technology, as a new treatment technology for antibiotic pollutants, is an efficient approach for the degradation of persistent organic pollutants and has great advantages in treatment effect and cost [1,14]. TiO${}_{2}$ is an outstanding representative of traditional photocatalysts due to its easy preparation, environmental impact and good stability [20]. However, the large band gap (around 3.2 eV) makes it only able to work under short wavelength ultraviolet light, while the utilization rate of visible light is quite poor. Unfortunately, ultraviolet light only accounts for about 5% of natural light [20]. This property makes TiO${}_{2}$ unable to effectively use the rich and cheap resource of natural light. Therefore, the development of TiO${}_{2}$-based catalysts that can effectively use visible light or full spectrum has always been research hot spots. In this case, narrowing the band gap of TiO${}_{2}$ by elemental doping to broaden the light response range and reduce the recombination of photogenerated electron-hole pairs seem to be an application trend [21,22]. Many previous works have proven that introducing the impurity metallic ions (or nonmetallic atom) into the lattice of TiO${}_{2}$ could replace Ti${}^{4}$${}^{+}$ (or lattice oxygen) and generate impurity energy levels, which could lower the Fermi level and trigger the optical absorption edge redshift changes [23,24]. In addition, with the partial substitution of impurity elements, the attendant defects would inevitably produce oxygen vacancies (OVs), further promoting the photocatalytic performance of TiO${}_{2}$ [22,25]. As a transition metal element with the same period as titanium, zirconium has the same outer electronic structure as titanium. Hence, zirconium could be an excellent candidate for doping TiO${}_{2}$.

Recently, employing natural minerals as the catalyst to build coupling systems has been emerging as a facile and effective way to enhance the catalyst performance with quite a low cost in comparison with other artificial synthesis supports [26,27]. Natural minerals, including diatomite [28,29,30], kaolinite [31,32,33], halloysite [34], etc., have the features of abundant pore structure and good stability and are environmentally friendly. Among them, natural diatomite is a siliceous biological sedimentary rock mainly composed of amorphous silica [35]. It is naturally endowed with a disk-shaped structure and a macroporous–mesoporous system, which makes it an excellent support candidate. Many previous studies have proven that the porous structure and the abundant hydroxyl surface of natural diatomite could not only improve the despersion and reduce the particle size of the catalysts, but also provide more adsorption reaction active sites for the photocatalytic process [25,30]. Moreover, its charged surface could also accelerate the separation of photogenerated electron-hole pairs, thus improving the photocatalytic properties. Hence, it is promising to integrate a TiO${}_{2}$-based catalyst and diatomite to construct a composite system with catalytic properties and dispersion.

In the current work, zirconium doped TiO${}_{2}$ was in situ synthesized on the surface of natural diatomite via a facile sol–gel process to improve the photocatalytic degradation of TC under visible light. The structural and photoelectrochemical properties of the as-prepared samples were systematically investigated and analyzed via a series of characterizations. The photocatalytic performance was evaluated by the photodegradation of TC in aqueous solution under visible light. The potential degradation intermediates and pathways, as well as the relevant enhanced mechanism, are also discussed and proposed in this work.

## 2. Materials and Methods

### 2.1. Chemicals and Materials

The natural diatomite (denoted as D) used in this work is from Linjiang city, Jilin province, China [25]. The reagents used in this work are all purchased from Shanghai Macklin Biochemical Co., Ltd (Shanghai, China), including acetic acid (CH${}_{3}$COOH, AR), ethanol absolute (C${}_{2}$H${}_{5}$OH, AR), tetrabutyl titanate (C${}_{16}$H${}_{36}$O${}_{4}$Ti, donated as TBOT, AR), tert-butanol (C${}_{4}$H${}_{10}$O, donated as t-BuOH, AR), ethylenediaminetetraacetic acid disodium (C${}_{10}$H${}_{14}$N${}_{2}$Na${}_{2}$O${}_{8}$, donated as EDTA-2Na, AR), hydrochloric acid (HCl, AR), silver nitrate (AgNO${}_{3}$, AR) and zirconium oxychloride octahydrate (ZrOCl${}_{2}$·8H${}_{2}$O, AR). Deionized water (≥10.0 MΩ cm) was used throughout our experiments.

### 2.2. Preparation of Zr Doped TiO${}_{2}$/Diatomite Photocatalysts

The Zr-doped TiO${}_{2}$/diatomite composite was prepared by a sol–gel process based on our previous reports [25]. In a typical process, 2 mL of CH${}_{3}$COOH, 3 mL of TBOT and appropriate amount of ZrOCl${}_{2}$·8H${}_{2}$O are successively added into 24 mL of anhydrous ethanol, which is designated as Solution A. Meanwhile, we prepare an ethanol aqueous solution (v(CH${}_{3}$CH${}_{2}$OH):v(H${}_{2}$O) = 1:1) with pH = 2 as Solution B. Then, Solution B is added dropwise to Solution A under strong stirring to form a homogeneous solution (desiginated as Solution C) at room temperature. Next, 1 g of D is dispersed into Solution C. After vigorous stirring for 1 h, we continue stirring for 10 h under mild stirring until the solution becomes gelatinous. In the following step, the obtained gel is dried at 80 °C overnight and subsequently ground into powder. The resultant powder is calcined at 500 °C for 2 h under air conditioning with a heating step of 5 °C·min${}^{-}$${}^{1}$. The obtained pale yellow powder is Zr-doped TiO${}_{2}$/diatomite (designated as ZrTD) composite. The proportions of Zr and TiO${}_{2}$ in the as-synthetized doping composites are adjusted by the mass ratios of r = m(Zr):m(TiO${}_{2}$), where r is set as 1%, 2%, 3% and 4%, respectively. Hence, the assynthetized doping composites are successively donated as 1%ZrTD, 2%ZrTD, 3%ZrTD and 4%ZrTD, respectively. The pure TiO${}_{2}$ (designated as T) and Zr-doped TiO${}_{2}$ (designated as ZrT) are also prepared based on the above procedure for comparison. In addition, the inductively coupled plasma-mass spectrometry (ICP-MS) results showed that the Zr ions in the 3% ZrTD composite could be detected as 1.86%, which indicates that the actual doping amount is about 1.86%.

### 2.3. Equipment and Characterizations

The X-ray diffraction patterns (XRD) are analyzed using a D8 Advance XRD with Cu Kα radiation (Bruker Corporation, Billerica, MA, USA). The morphologies are carried out with the field-emission scanning electron microscope (FESEM) with energy dispersive spectrometer (EDS) (Hitachi SU-8020, Hitachi, Ltd., Tokyo, Japan) and high-resolution transmission electron microscope (HR-TEM) (Tecnai G2 F20, FEI company, Hillsboro, OR, USA operating at 200 kV, USA). The UV-vis diffuse reflectance spectra (UV-vis DRS) are performed on a Shimadzu UV2550 UV-vis spectrophotometer (SHIMADZU Corporation, Kyoto, Japan). A Thermo ESCALAB 250Xi with Al Kα X-ray radiation (hν = 1486.6 eV) (Thermo Fisher Scientific, Waltham, MA, USA) is applied to obtain the X-ray photoelectron spectrometer (XPS). The electrochemistry properties including the photocurrent (i-t) and electrochemical impedance spectroscopy (EIS) are recorded by a CHI-660B electrochemical analyzer (CH Instruments Inc., Austin, TX, USA). An FLS920 life and steady state spectrometer at room temperature with the excitation wavelength at 465 nm (Edinburgh Instruments Ltd., Edinburgh, UK) is employed to test the photoluminescence (PL) spectra. The reductive pathway and intermediates were identified by using a High-Performance Liquid Chromatography-Mass Spectrometry (HPLC-MS) (Bruker, Germany). The Zr ions amount were obtained on a Agilent 7700 inductively coupled plasma-mass spectrometry (ICP-MS) (Agilent Technologies Inc., Palo Alto, CA, USA).

### 2.4. Evaluation of Photocatalytic Activity

The tetracycline (TC) is selected as target pollutant to evaluate the photocatalytic activities of the as-prepared samples. The photocatalytic degradation process is conducted under a 500 W Xenon lamp (BL-GHX-V, Shanghai Bilang Plant, Shanghai, China) with a 420 nm cut-off filter. In a typical process, 100 mg of as-prepared samples is first mixed with 100 mL of TC aqueous solution (20 mg·L${}^{-}$${}^{1}$), respectively, and then ultrasonic treatment is carried out for 15 min to obtain better dispersion. Prior to the light reaction process, the equilibrium of adsorption–desorption is achieved via 60 min of reaction in a dark condition. After conducting under a light condition, 3 mL of suspension is taken out at a specific time interval, and the solid–liquid separation is realized through the 0.22 μm water filter membrane. The obtained solution is further measured at 358 nm by the UV-Vis spectrophotometer. The degradation ratio curves of TC are calculated by C${}_{t}$/C${}_{-}$${}_{30}$ of the as-prepared samples, where C${}_{t}$ is the concentration of TC at time t and C${}_{-}$${}_{30}$ is the initial concentration of TC.

## 3. Results and Discussions

Figure 1 shows the SEM images of D and 3%ZrTD composite. As shown in Figure 1a, the natural diatomite used in this work is in the shape of a disk with a diameter of about 20 μm, and penetrating pores with a diameter of about 500 nm are regularly distributed on the surface. Its flat and porous surface structure is an excellent candidate as a support for catalyst [25]. Revealed in Figure 1b, the diatomite surface of 3%ZrTD composite becomes rough and full of spherical nanoparticles with a diameter of about several tens of nanometers. The spherical particles are TiO${}_{2}$ nanoparticles, and they are uniformly dispersed on the surface of diatomite and combined with the surface. The uniform dispersion with smaller particle size could increase the adsorption reaction active sites, thus improving the photocatalytic performance. To reveal the ingredients of the 3%ZrTD composite, moreover, the SEM-EDS element mapping is employed to investigate the sample. As exhibited in Figure 1c, Zr, Si and Ti elements are observed within the diatomite disk area, and the elements show uniform distribution in the composite, which solidly certifies that Zr is doped into the TiO${}_{2}$, and the Zr-doped TiO${}_{2}$/diatomite catalytic system is successfully established. In particular, there also exists TiO${}_{2}$ formed as individual particles, which might be attributed to the synthesis procedure: the addition of Solution B in Solution A before the introduction of diatomite allows hydrolysis of the alkoxides before the precursors loading on diatomite. Furthermore, to reveal the effect of Zr ion doping on the lattice structure of TiO${}_{2}$, the HRTEM is carried out on the 3%ZrTD composite (Figure 1d). The results show that its lattice spacing is 0.350 nm, which is consistent with the (101) crystal plane of anatase phase. The crystalline phase of the as-prepared samples is revealed by XRD analysis in Figure 1d. In the XRD pattern of D, the diffraction peak at 2θ = 26.63° demonstrates the existence of quartz impurity [36], while the broad band between 2θ = 17.0° to 26.0° is attributed to the amorphous silica (SiO${}_{2}$). For the other samples, the characteristic diffraction peaks at 2θ = 25.19° (101), 37.70° (004), 48.02° (200) and 54.97° (211) are consistent with the corresponding diffraction planes of anatase phase TiO${}_{2}$ (JCPDS 21-1272), and no other characteristic peaks are observed. It is indicated that the anatase phase with high photocatalytic activity is still retained in as-prepared TiO${}_{2}$ after the introduction of Zr ions, and no other possible impurities are generated. In addition, XPS is carried out to ascertain the surface chemical properties of the as-prepared 3%ZrTD composite. In detail, the high resolution XPS spectra of Zr, Ti, Si and O are displayed in Figure 1f–i, respectively. As shown in Figure 1f, the apparent peaks at 181.55 eV and 183.84 eV belong to Zr${}^{4}$${}^{+}$ 2p${}_{3}$${}_{/}$${}_{2}$ and Zr${}^{4}$${}^{+}$ 2p${}_{1}$${}_{/}$${}_{2}$, respectively [23]. It is proved that Zr ions are successfully doped into the crystal lattice of TiO${}_{2}$ and exist in Zr${}^{4}$${}^{+}$ form. Figure 1g is a high-resolution XPS spectrum of Ti 2p, and the peaks at around 458.43 eV, 459.55 eV, 463.93 eV and 464.87 eV can be allocated to Ti${}^{4}$${}^{+}$ 2p${}_{3}$${}_{/}$${}_{2}$, Ti${}^{3}$${}^{+}$ 2p${}_{3}$${}_{/}$${}_{2}$, Ti${}^{4}$${}^{+}$ 2p${}_{1}$${}_{/}$${}_{2}$ and Ti${}^{3}$${}^{+}$ 2p${}_{1}$${}_{/}$${}_{2}$, respectively [25]. The existence of Ti${}^{3}$${}^{+}$ indicates that the introduction of Zr ions leads to distortion in the lattice of TiO${}_{2}$, resulting in a defect structure, which could be conducive to improving the visible light response [37]. Figure 1h displays Si 2p XPS spectrum. The peaks situated at 103.00 eV and 103.88 eV are the characteristic peaks of Si 2p of SiO${}_{2}$ and Si 2p of Ti/SiO${}_{2}$, resulting in the bonding relationship between diatomite and TiO${}_{2}$ via the Ti–Si bonds [30]. In addition, as shown in the O 1s spectrum (Figure 1i), five obvious characteristic peaks are assigned to Ti–O bonds of typical lattice oxygen (530.09 eV), surface hydroxyl (530.88 eV), surface O defects (532.18 eV), adsorbed H${}_{2}$O (532.88 eV) and Si–O bonds of SiO${}_{2}$ (533.25 eV), respectively. The high surface hydroxyl and surface O defects level are beneficial for the transfer and migration of e${}^{-}$-h${}^{+}$ pairs, which are attributed to the doping of Zr ions [37]. Combining with the results of SEM-EDS and XPS analysis, it could powerfully prove that Zr has been successfully doped into TiO${}_{2}$, and the adsorption–degradation collaborative system has also been established through the intimate combination between the TiO${}_{2}$ and natural diatomite.

Figure 2 illustrates the UV–Vis DRS and the corresponding Kubelka–Munk function curves of the as-prepared samples. The optical properties can be seen from Figure 2a. T and TD composite only have powerful absorbance in the UV region, where the light adsorption edges are both at around 385 nm. After doping with Zr ions, in contrast, the light adsorption edges of ZrT and 3%ZrTD composite are redshifted at around 415 nm. This can indicate that the absorption capacity of ZrT and ZrTD composite to visible light is enhanced after Zr ions doping, which makes it possible to have a visible light response and convert more visible light energy into chemical energy. In addition, the band gaps of the as-prepared samples are revealed from the Kubelka–Munk function curves in Figure 2b. The optical band gap values of T, TD composite, ZrT and 3%ZrTD composite are estimated to be 3.19 eV, 3.19 eV, 3.07 eV and 3.04 eV, respectively. The band gap of T and the TD composite are lower than that of the ZrT and the 3%ZrTD composite, indicating that the band gap of pure TiO${}_{2}$ is reduced via doping with Zr ions. The lower band gap values could reduce the Fermi energy level of the catalyst, thus reducing the energy of photogenerated electrons migrating from the valence band to the conduction band. As a result, the separation efficiency of photogenerated electron-hole pairs as well as the utilization of visible light energy could be remarkably improved.

Photocatalytic behaviors of the as-prepared samples under visible light are evaluated by the photocatalytic degradation of the TC aqueous solution (20 mg·L${}^{-1}$) manifested in Figure 3. The photolysis of the TC under visible light irradiation is negligible. Clearly, as shown in Figure 3a, the equilibrium of adsorption–desorption between all the as-prepared samples and TC can be well-established in 30 min. After 180 min of visible light irradiation, D displays insignificant photocatalytic activity. The removal rates of both the T and the TD composite towards TC are poor, due to the little visible light response of pure TiO${}_{2}$. In comparative with them, the removal rates of the samples doped with Zr ions are significant improved, in which 3%ZrTD composite presents the best catalytic property (reach to 95.32%). Hence, the optimal doping amount of Zr ion is 3%. Excessive Zr ions would increase the recombination centers of photogenerated carriers and be harmful to the separation of photogenerated electron-hole pairs, thus reducing the photocatalytic performance. Interestingly, it is clear that introducing the natural diatomite as support improves the photocatalytic performance of the catalyst, which is consistent with our expectations. In addition, as shown in Figure 3b, the photocatalytic process of TC over the as-prepared samples can be well described by the first-order kinetic equation, ln(C${}_{0}$/$C_{t}$) = kt (R${}^{2}$ > 0.98, Table A1). In addition, the corresponding reaction rate constant (the parameter k in the equation) of TC is also provided in Figure 3b, and the details of kinetic parameters are listed in Table A1. The reaction rate constants (k) of 3%ZrTD composite is much higher than other comparative samples which is calculated to be 1.901 × 10${}^{-}$${}^{2}$ min${}^{-}$${}^{1}$, which is around 13.53 times and 8.65 times higher than that of the as-prepared T (0.718 × 10${}^{-}$${}^{2}$ min${}^{-}$${}^{1}$) and ZrT (1.179 × 10${}^{-2}$ min${}^{-1}$), respectively, and it is relatively high compared with other referred samples (Table A2). The outcomes confirm that doping appropriate Zr ions into TiO${}_{2}$ is an effective and facile way to enhance its visible light response catalytic activity. Furthermore, the fresh 3%ZrTD composite and the used one are measured by XRD pattern (Figure 3c). The results reveal that the crystal structure hardly changes before/after the photocatalytic process, suggesting the excellent texture stability. Moreover, the radical scavenger experiments are carried out to reveal the major reactive oxygen species (ROS) that are involved in the photocatalytic degradation process of the 3%ZrTD composite. In this work, as shown in Figure 3d, we add the scavengers of AgNO${}_{3}$, EDTA-2Na, t-BuOH and BQ to catch electron (e${}^{-}$), hole (h${}^{+}$), hydroxyl (•OH) radicals and superoxide (•O${}_{2}$${}^{-}$) radicals, respectively. The results show that the photocatalytic degradation performance of TC is restrained after the addition of scavengers into the photocatalytic system. AgNO${}_{3}$ and t-BuOH have the almost no inhibitory effect, which around 91.29% and 89.68% of TC can be removed, respectively. In addition, around 52.26% and 26.61% of TC can be removed via adding EDTA-2Na and BQ as scavengers, respectively. The results indicate that photogenerated h${}^{+}$ and •O${}_{2}^{-}$ radicals play the vital role in the photocatalytic process, that is, they are the major ROS radicals during the photocatalytic process.

The photocurrents of the T, ZrT and 3%ZrTD composite are tested by turning the visible light on/off with a 30 s interval. As demonstrated in Figure 4a, the 3%ZrTD composite has higher photocurrent signals than the other two samples, which further certifies that the 3%ZrTD composite has higher separation efficiency of electron-hole pairs after Zr ions doping. As shown in Figure 4b, the equivalent circuit diagram is shown by fitting the EIS plots in which R${}_{e}$ is the resistance of electrolyte solution; R${}_{c}$${}_{t}$ is the charge-transfer resistance; Z${}_{W}$ is the Warburg impedance; and Q${}_{d}$${}_{l}$ is the constant phase element of electrode/electrolyte interface. Based on the Nyquist circle principle, the smaller arc diameter indicates lower resistance (14.08Ω), which is also favorable to the transfer of electron-hole pairs. On the other hand, the PL analysis, as displayed in Figure 4c, is tested to better understand the separation and recombination of carriers of the as-prepared samples, including T, ZrT and 3%ZrTD composite. Clearly, the three photocatalytic systems produce obvious PL signals in the range from 400 nm to 500 nm. Compared with the other two samples, the PL intensity of 3%ZrTD composite is relatively low, suggesting that the electron migration efficiency is enhanced after doping with Zr ions. In general, the higher efficiency of the electron migration means the stronger photocatalytic activity. In addition, the fluorescence peaks of the samples with doping with Zr ions have a slight blue shift, resulting from the enhancement of the visible light response. Combined with PL analysis and electrochemical study results, it could be reasonably determined that the Zr-doped TiO${}_{2}$ with diatomite as the support manifests a noteworthy action in improving the separation of electron-hole pairs.

To identify the transformation pathway of TC in the presence of the 3%ZrTD composite under visible light irradiation, the main intermediate products in the photodegradation process are investigated and revealed by HPLC-MS analysis. The corresponding mass spectra after reacting for 120 min are displayed in Figure 5. Based on the mass-to-charge ratios (m/z) obtained from HPLC-MS analysis in Figure 5, a possible potential transformation (degradation) pathway could be proposed as displayed in Figure 6 [38,39,40,41]. Specifically, TC (P1, m/z = 444.93) is first oxidized to oxytetracycline (OTC, P2, m/z = 457.09), where it forms a ketone at position 13. Then, the methyl in the amino groups is bi-demethylated due to the low bond energy of C–N and forming P3 (m/z = 431.16). Subsequently, the deamination reaction occurs in the P3 to form the P4 (m/z = 401.08). After that, the P5 (m/z = 385.12) is generated by removing the ketone of the P4 at position 3. In the following transformation process, the P4 successively removes ketone, hydroxyl and methyl and finally forms P8 (m/z = 293.21), which goes through P5 (m/z = 385.12), P6 (m/z = 357.24) and P7 (m/z = 309.11). After that, the ring D of P8 begins to break from positions 5 and 14 leading to the generation of P9 (m/z = 224.09). With further oxidization, the P9 has three conversion paths. For Pathway 1 (P9→P10→P12→P14), the ring C of P9 is directly removed by oxidization at positions 7 and 12 to obtain P10 (m/z = 146.61). In addition, the P10 subsequently oxidizes at positions 9 and 10 to remove the ring B and obtain P12 (m/z = 129.80). Finally, the P12 opens the ring at position 10 and is finally oxidized to small molecules via P14 (m/z = 113.11). Pathway 2 (P9→P11→P10→P12→P14) is similar with Pathway 1. It just goes through the P12 (m/z = 174.96) in the process from P9 to P11, which could be the refinement of Pathway 1. Pathway 3 (P9→P11→P13→P12→P14) is a variant of the first two transformation paths. It opens the ring in turn rather than directly removing the ring in which P11 is opened at the double bond at position 12 and converted to P12 via P13 (m/z = 180.98). As a result, those intermediate products are degraded and evolved to CO${}_{2}$, H${}_{2}$O, and NH${}_{4}$${}^{+}$. Based on the results and analysis of HPLC-MS for TC degradation, it is concluded that the TC molecule could be degraded to small molecules under the action of the ZrTD composite and the visible light illumination.

In summary, based on the previous characterization and analysis, the visible light enhancement mechanism of the ZrTD composite and the corresponding photocatalytic degradation process of TC can be proposed as shown in Figure 7. Specifically, after Zr ions doping, a new energy level of Zr 3d (impurity level) could be generated between the conduction band (CB) and the valance band (VB) in the band structure of TiO${}_{2}$. In general, the impurity level could be used as a springboard for electron transition from VB to CB, thus improving the separation efficiency of photogenerated carriers. On the other hand, the doping of Zr ions leads to the generation of lattice defects in TiO${}_{2}$ (O defects), resulting in the formation of surface oxygen vacancies (OVs), which is proved by the XPS analysis. The existence of the surface OVs could generate a donor energy level below the CB of TiO${}_{2}$, thus further reducing the energy barrier of electron transition and broadening the light response to the visible range. Combined with the above two points, the apparent phenomenon is that the band gap of the composite is reduced after doping of Zr ions, and the photocatalytic performance of the composite is significantly enhanced under visible light. In addition, the introduction of the natural diatomite also plays a vital role in enhancing the photocatalytic performance of the composite. The porous structure and the abundant hydroxyl surface of the natural diatomite cannot only improve the dispersion and reduce the particle size of the TiO${}_{2}$, but also provide more adsorption reaction active sites for the photocatalytic process. Moreover, its charged surface could also accelerate the separation of photogenerated electron-hole pairs. Overall, they form an adsorption–degradation collaborative system to efficiently degrade pollutants. The photocatalytic degradation process could be inferred as follows. The TC molecules are first adsorbed onto the surface of the ZrTD composite. Then, under visible light illumination, the electrons in the VB of TiO${}_{2}$ would first transfer into the Zr 3d energy level, and then transfer into the CB of TiO${}_{2}$. The holes left in VB directly react with TC molecules, while the electrons in the CB combine with oxygen molecules in the air to form •O${}_{2}$${}^{-}$ radicals. As a result, the generated h^+^ and •O${}_{2}$${}^{-}$ radicals with strong oxidation capacity can oxidize TC molecules into small molecules, including CO${}_{2}$, H${}_{2}$O and NH${}_{4}$${}^{+}$, et al.

## 4. Conclusions

In summary, a novel zirconium-doped TiO${}_{2}$/diatomite (ZrTD) composite was successfully synthesized through a facile sol- gel process, and the photocatalytic performance under visible light was evaluated by the degradation of tetracycline TC under visible light irradiation. The optimal doping ratio of zirconium into TiO${}_{2}$ was obtained at 3% (3%ZrTD composite), whose degradation rate constant for TC is up to around 13.53 times and 8.65 times higher that of pure TiO${}_{2}$ and zirconium-doped TiO${}_{2}$, respectively. The improvement of photocatalytic performance could be mainly attributed to zirconium ions doping and the introduction of diatomite support. The zirconium doping could introduce the impurity levels of Zr 3d and oxygen vacancies into the lattice of TiO${}_{2}$, resulting in broadening the light absorption range, reducing the band gap and improving the separation efficiency of photogenerated electron-hole pairs. On the other hand, the introduction of diatomite could provide more adsorption reaction active sites for the photocatalytic process. In addition, the main active species worked during the degradation process of TC were the photogenerated holes (h${}^{+}$) and superoxide (•O${}_{2}$${}^{-}$) radicals. Our work would like to give a new perspective on developing mineral-based photocatalysts with visible light response.

## Figures and Tables

**Figure 1 nanomaterials-12-02827-f001:**
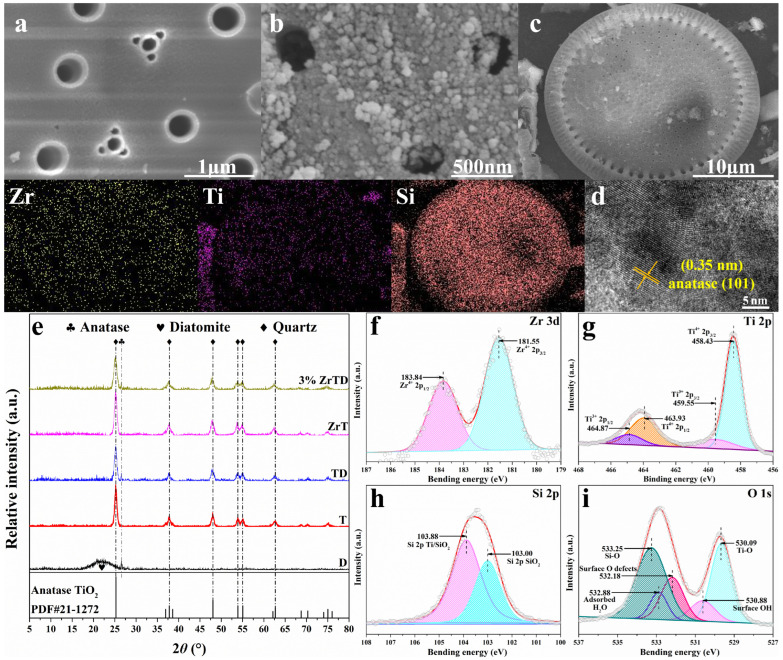
SEM images of: (**a**) D; (**b**) 3%ZrTD composite; (**c**) SEM-EDS element mapping of Zr, Ti and Si for 3%ZrTD composite, and the upper left corner shows the corresponding images; (**d**) HRTEM images of 3%ZrTD composite; (**e**) XRD patterns of D, T, TD composite, ZrT and 3%ZrTD composite; XPS spectra of (**f**) Ti 2p; (**g**) Si 2p; (**h**) Zr 3d; and (**i**) O 1s of 3%ZrTD composite.

**Figure 2 nanomaterials-12-02827-f002:**
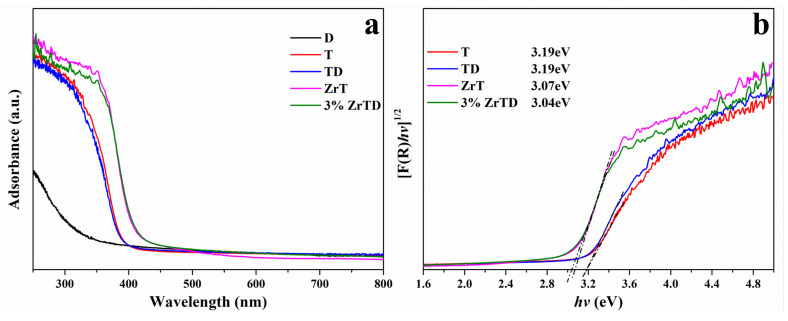
(**a**) UV-Vis DRS; and (**b**) band gaps of D, T, TD composite, ZrT and 3%ZrTD composite.

**Figure 3 nanomaterials-12-02827-f003:**
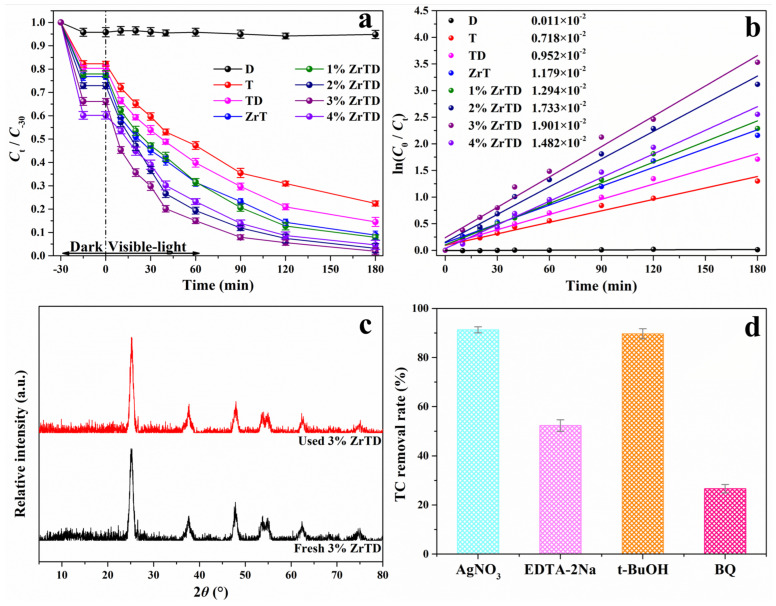
(**a**) Photocatalytic removal curves of TC in the presence of D, T, TD composite, ZrT and ZrTD composites with different Zr doping amounts under visible light irradiation; (**b**) the corresponding first-order kinetics plots with the photocatalytic reaction rate constant values; (**c**) XRD patterns of fresh and used 3%ZrTD composite; and (**d**) radical scavenger experiments under visible light of 3%ZrTD composite towards TC.

**Figure 4 nanomaterials-12-02827-f004:**
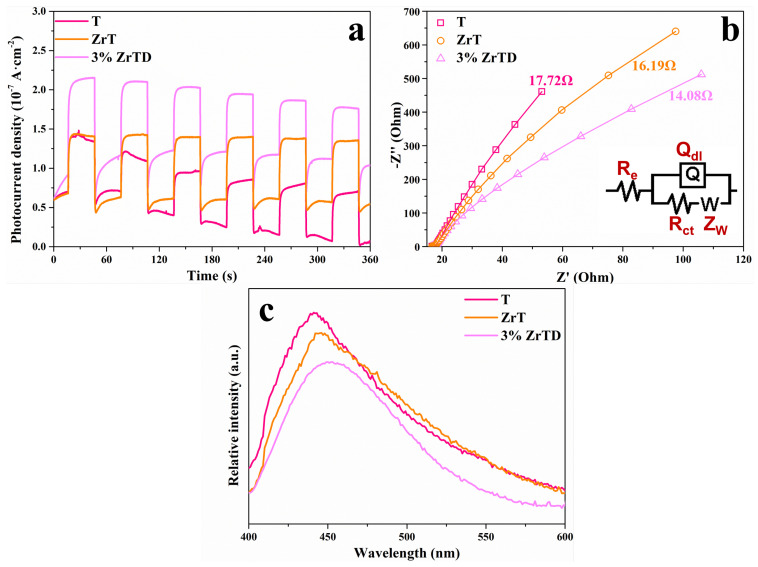
(**a**) photocurrent curves (i–t); (**b**) EIS plots; and (**c**) PL spectra of T, ZrT and 3%ZrTD composite.

**Figure 5 nanomaterials-12-02827-f005:**
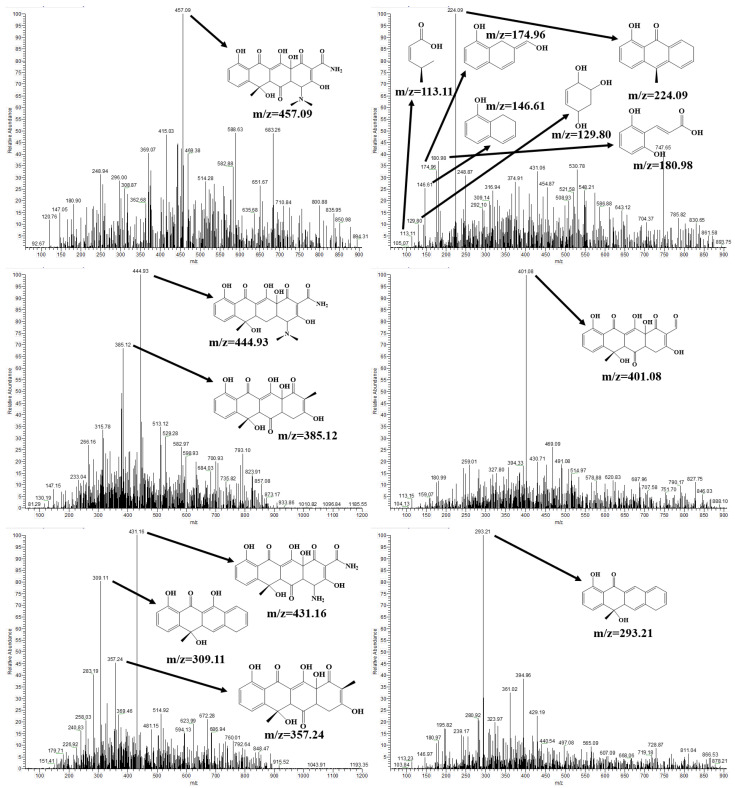
HPLC-MS spectra of aqueous system in photocatalytic reaction after 120 min.

**Figure 6 nanomaterials-12-02827-f006:**
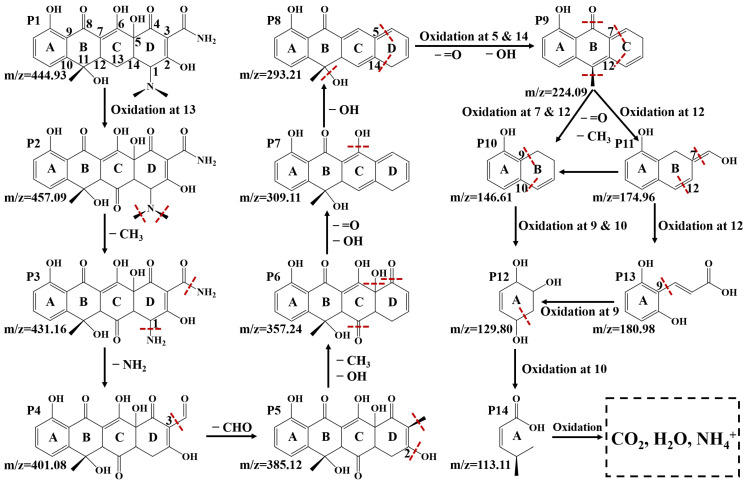
Proposed possible degradation pathway and main intermediates for TC in the presence of the 3%ZrTD composite under visible light irradiation.

**Figure 7 nanomaterials-12-02827-f007:**
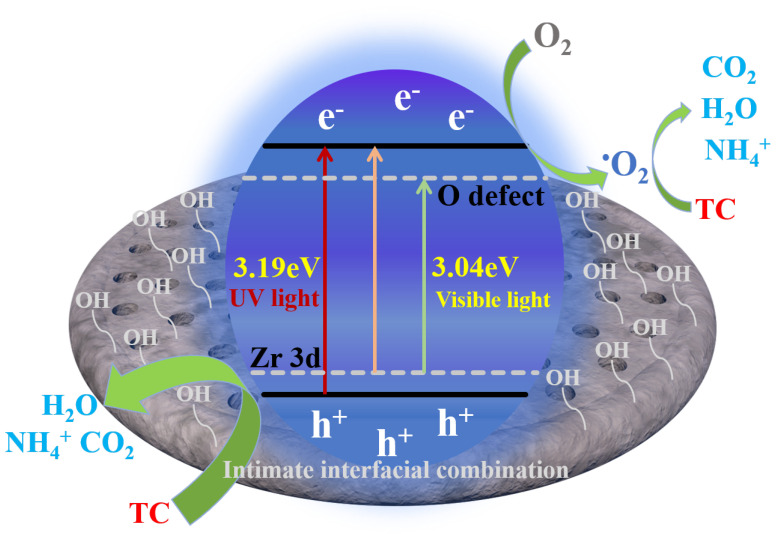
Diagram description of the photocatalytic performance enhancement mechanism for the ZrTD composite.

## Data Availability

Data are contained within the article.

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
