# Peer review of "Natural Diatomite Supported Zirconium-Doped TiO2 with Tailoring Band Structure for Enhanced Visible-Light Photocatalytic Properties"

_nanomaterials, 2022, doi:10.3390/nano12162827_

Round 1

Reviewer 1 Report

The research article on " Natural diatomite supported zirconium doped TiO2 with tailoring band structure for enhanced visible-light photocatalytic properties" reports the synthesis of Zr loaded TiO2 showing photocatalytic degradation of tetracycline. Below are some of the comments that authors should look into improving the manuscript.

  1. It would be better if the authors could provide more information about the d-spacing and explain how it correlates to the property.
  2.  The impurity peaks (XRD) mentioned in the figure and text are not matching?
  3. Better to quantify the amount of Zirconium using ICP-MS.
  4. The XPS deconvolutions should be refixed as the experimental and fitting data are not properly fitted with the proper background.
  5. Why is there a huge decrease in the binding energy due to zirconium doping into TiO2? As the binding energy of ZrO2 is higher (5.0 eV) than TiO2 (3.2 eV), a doping amount of only 3%  doping will not show such a change. Better to re-check the UV-Vis spectra.
  6. Figure 3 shows photocatalytic removal of 1%, 2%, and 4% ZrTD samples, but no other information about the material characterization in the manuscript? better to include.
  7. First-order kinetics doesn’t fit well for the samples. What could be the reason? And add information on the fitting parameters as a table.
  8. A table showing the photocatalytic degradation of TC in comparison to the best material should be included and explained.
  9. EIS plots of the samples are shown in Figure 4. Better to explain it more with the resistance values.
  10. The authors should check for mistakes, Line 140 (mentioned 20 m) scale bar of SEM image.
  11. Add the y-axis scale in Figure 4b EIS plots.
  12. What is the novelty of the work compared to Zr-doped TiO2 catalyst used for photocatalytic degradation of tetracycline?
  1.  

Author Response

Response to Reviewer #1

Reviewer #1: The research article on "Natural diatomite supported zirconium doped TiO2 with tailoring band structure for enhanced visible-light photocatalytic properties" reports the synthesis of Zr loaded TiO2 showing photocatalytic degradation of tetracycline. Below are some of the comments that authors should look into improving the manuscript.

Comment 1:

It would be better if the authors could provide more information about the d-spacing and explain how it correlates to the property.

Response: Thanks for the reviewer’s valuable comment. After receiving the comments from the reviewer, we carried out high-resolution transmission microscope (HRTEM) on the 3%ZrTD composite (Figure 1d). The results show that its lattice spacing is 0.350 nm, which is consistent with the (101) crystal plane of anatase phase. Consequently, the Zr ion doping will not affect the lattice structure of TiO2 in the composite samples prepared by our proposed preparation process. The reason might be that the doping amount is too small. In addition, we also found from the relevant reports (ref: Ind. Eng. Chem. Res. 2018, 57, 14044−14051) that the Zr/TiO2 composite with low doping amount will not affect the lattice structure of TiO2. Please check the revised manuscript. The revised contents are as follow:

Furthermore, in order to reveal the effect of Zr ion doping on the lattice structure of TiO2, the HRTEM is carried out on 3%ZrTD composite (Figure 1d). The results show that its lattice spacing is 0.350 nm, which is consistent with the (101) crystal plane of anatase phase.

Figure 1. SEM images of (a) D, (b) 3%ZrTD composite, (c) SEM-EDS element mapping of Zr, Ti and Si for 3%ZrTD composite, and the upper left corner shows the corresponding images; (d) HRTEM images of 3%ZrTD composite; (e) XRD patterns of D, T, TD composite, ZrT composite and 3%ZrTD composite; XPS spectra of (f) Ti 2p, (g) Si 2p, (h) Zr 3d and (i) O 1s of 3%ZrTD composite.

Comment 2:

The impurity peaks (XRD) mentioned in the figure and text are not matching?

Response: Thanks for the reviewer’s valuable comment. The impurity peaks in the XRD patterns of the samples in our work are originated from the quartz in diatomite. After receiving the reviewers' comments, we carefully checked the figure and the corresponding text. We found that the quartz impurity is located at 2θ = 26.63°, which is consistent with previous reports (ref: Build. Environ. 219 (2022) 109216; Sep. Purif. Technol. 297 (2022) 121477). Please check the revised manuscript.

Comment 3:

Better to quantify the amount of Zirconium using ICP-MS.

Response: Thanks for the reviewer’s valuable comment. According to the suggestion, we have supplemented the ICP-MS to quantify the amount of Zirconium Please check the revised manuscript. The corresponding text have been revised as follows:

In addition, the ICP-MS results showed that the Zr ions in the 3%ZrTD composite could be detected as 1.86%, which indicates that the actual doping amount is about 1.86%.

Comment 4:

The XPS deconvolutions should be refixed as the experimental and fitting data are not properly fitted with the proper background.

Response: Thanks for the reviewer’s valuable comment. According to the suggestions of the reviewer, we tried to refix the XPS deconvolution using different backgrounds, including Linear, Shirley and Tougaard types. The results show that the deconvolution with linear background is the most suitable. The revised results were put into the revised manuscript. Please check the revised manuscript.

Comment 5:

Why is there a huge decrease in the binding energy due to zirconium doping into TiO2? As the binding energy of ZrO2 is higher (5.0 eV) than TiO2 (3.2 eV), a doping amount of only 3% doping will not show such a change. Better to re-check the UV-Vis spectra.

Response: Thanks for the reviewer’s valuable comment. We retested the UV-Vis spectrum of the 3%ZrTD composite, and obtained the same result as those in the manuscript. In addition, we have reviewed the relevant previous reports, and the band gap of the Zr doped TiO2 in the previous reports is also very close to the samples prepared in our work (ref: Ind. Eng. Chem. Res. 2018, 57, 14044−14051). The reason for the decrease in the binding energy of the composite could not be attributed to the formation of the ZrO2, but the introduction of the impurity level and oxygen defects, and further broadening the light response (ref: Ind. Eng. Chem. Res. 2018, 57, 14044−14051). Hence, the zirconium elements doping into TiO2 makes the binding energy decrease.

Comment 6:

Figure 3 shows photocatalytic removal of 1%, 2%, and 4% ZrTD samples, but no other information about the material characterization in the manuscript? better to include.

Response: Thanks for the reviewer’s valuable comment. We are sorry that we did not explain clearly the role of these samples included in this experiment. This part of experiment is aim to reveal the optimal doping amounts of the Zr elements, and the results show that the optimal doping amounts is 3%. These three comparative samples have achieved this goal here. On the other hand, the structural characteristics 3%ZrTD composite is the most representative and can best reflect the influence of Zr doping on the structure and properties of materials. Hence, selecting 3%ZrTD sample for characterization and research can help readers quickly and intuitively see the significance of doping Zr.

Comment 7:

First-order kinetics doesn’t fit well for the samples. What could be the reason? And add information on the fitting parameters as a table.

Response: Thanks for the reviewer’s valuable comment. As suggested by the reviewer, we have supplemented fitting parameters as a table (Table A1). From the Table A1, we could find that the values of fitting constant R-squared (R2) of these samples are all greater than 0.98, indicating that the pseudo-first-order kinetics fitted well for the photocatalytic degradation process of the as-prepared samples. Please check the revised manuscript. The Table A1 is as follows:

Table A1.

Pseudo-first order kinetic parameters for the photocatalytic degradation of TC by T, TD, ZrT, 1%ZrTD, 2%ZrTD, 3%ZrTD and 4%ZrTD

Sample

k1

 (min-1)

R2

T

0.718×10-2

0.9889

TD

0.952×10-2

0.9854

ZrT

1.179×10-2

0.9849

1%ZrTD

1.294×10-2

0.9855

2%ZrTD

1.733×10-2

0.9846

3%ZrTD

1.901×10-2

0.9889

4%ZrTD

1.482×10-2

0.9825

Comment 8:

A table showing the photocatalytic degradation of TC in comparison to the best material should be included and explained.

Response: Thanks for the reviewer’s valuable comment. As suggested by the reviewer, we have given a comparative table about the latest TC degradation results based on photocatalytic techniques. Seen from Table A2, it is indicated that the prepared composite materials in this study exhibits better performance than the comparative reports. Please check the revised manuscript. The comparative Table A2 is listed as follow:

Table A2.

The latest TC degradation results based on different photocatalytic systems under visible light illumination.

Photocatalysts

Photocatalyst concentration

(g·L-1)

TC concentration

(mg·L-1)

Removal

time

(min)

Removal

rate

(%)

References

3%ZrTD

1.0

20

120

93.5

Present work

P25

0.1

10

90

29.1

[1]

Bi2WO6

1.0

20

120

48.4

[2]

PO43--Bi2WO6/ polyimide

1.0

20

120

65.1

[2]

MoS2

0.2

20

180

48.5

[3]

MoS2@Z-5

0.5

10

180

87.2

[3]

CuInS2/Bi2MoO6

0.6

15

120

84.7

[4]

Ag/g-C3N4-Ag-Ag3PO4

1.0

10

100

90.2

[5]

ZnO

0.1

50

240

83.7

[6]

MIL-100(Fe)

0.7

10

120

90.3

[7]

Zeolite@ZIF-67@PMS

-

50

60

93.7

[8]

Comment 9:

EIS plots of the samples are shown in Figure 4. Better to explain it more with the resistance values.

Response: Thanks for the reviewer’s valuable comment. As suggested by the reviewer, the equivalent circuit diagram and the transmission impedance (Rct) and solution impedance (Re) of electrons in the nanocomposite materials were calculated and given in Figure 4b. Please check the revised manuscript. The revised contents are as follows:

As shown in Figure 4b, the equivalent circuit diagram is shown by fitting the EIS plots, in which Re is the resistance of electrolyte solution; Rct is the charge-transfer resistance; ZW is the Warburg impedance; Qdl is the constant phase element of electrode/electrolyte interface. Based on the Nyquist circle principle, the smaller arc diameter indicates lower resistance (14.08Ω), which is also favourable to the transfer of electron-hole pairs.

Figure 4. (a) photocurrent curves (i-t) and (b) EIS plots (c) PL spectra of T, ZrT composite and 3%ZrTD composite.

Comment 10:

The authors should check for mistakes, Line 140 (mentioned 20 m) scale bar of SEM image.

Response: Thanks for the reviewer’s valuable comment. This is an error in the typesetting process using LaTeX. We are sorry for this careless mistake and have corrected it to μm in the revised manuscript. Please check the revised manuscript. The revised content is as follow:

As shown in Figure 1a, the natural diatomite used in this work is in the shape of a disk with a diameter of about 20 μm, and penetrating pores with a diameter of about 500 nm are regularly distributed on the surface.

Comment 11:

Add the y-axis scale in Figure 4b EIS plots.

Response: Thanks for the reviewer’s valuable comment. As suggested by the reviewer, we have added the y-axis scale in Figure 4b EIS plots. Please check the revised manuscript. The revised Figure 4 is as follow:

Figure 4. (a) photocurrent curves (i-t) and (b) EIS plots (c) PL spectra of T, ZrT composite and 3%ZrTD composite.

Comment 12:

What is the novelty of the work compared to Zr-doped TiO2 catalyst used for photocatalytic degradation of tetracycline?

Response: Thanks for the reviewer’s valuable comment. The main innovation of this work lies in the introduction of natural diatomite as a support, and constructing an adsorption-catalysis collaborative system. After introducing the natural diatomite, the photocatalytic performance is improved from three aspects:(1) Dispersion: improving the dispersion of the catalysts attributed to unique porous structure and the abundant hydroxyl surface; (2) Adsorption: providing more adsorption-reaction active sites for photocatalytic process; (3) Photocatalysis: accelerating the migration and separation of photogenerated carriers due to the charged surface. On the other hand, as far as we know, there is no previous reports focused on the Zr-doped TiO2 catalyst used for photocatalytic degradation of tetracycline.

Reference

[1] J.L. Tian, L.X. Wei, Z.Q. Ren, J.F. Lu, J. Ma, The facile fabrication of Z-scheme Bi2WO6-P25 heterojunction with enhanced photodegradation of antibiotics under visible light, J. Environ. Chem. Eng., 9 (2021) 106167.

[2] X. Gao, J. Niu, Y.F. Wang, Y. Ji, Y.L. Zhang, Solar photocatalytic abatement of tetracycline over phosphate oxoanion decorated Bi2WO6/polyimide composites, J. Hazard. Mater., 403 (2021) 123860.

[3] J.F. Liu, H. Lin, Y.B. Dong, Y.H. He, C.J. Liu, MoS2 nanosheets loaded on collapsed structure zeolite as a hydrophilic and efficient photocatalyst for tetracycline degradation and synergistic mechanism, Chemosphere, 287 (2022) 132211.

[4] J.R. Guo, L.P. Wang, X. Wei, Z.A. Alothman, M.D. Albaqami, V. Malgras, Y. Yamauchi, Y.Q. Kang, M.Q. Wang, W.S. Guan, X.T. Xu, Direct Z-scheme CuInS2/Bi2MoO6 heterostructure for enhanced photocatalytic degradation of tetracycline under visible light, J. Hazard. Mater., 415 (2021) 125591.

[5] S.Y. Li, M. Zhang, Z.H. Qu, X. Cui, Z.Y. Liu, C.C. Piao, S.G. Li, J. Wang, Y.T. Song, Fabrication of highly active Z-scheme Ag/g-C3N4-Ag-Ag3PO4 (110) photocatalyst photocatalyst for visible light photocatalytic degradation of levofloxacin with simultaneous hydrogen production, Chem. Eng. J., 382 (2020) 122394.

[6] P. Chen, N.N. Dong, J.J. Zhang, W. Wang, F.T. Tan, X.Y. Wang, X.L. Qiao, P.K. Wong, Investigation on visible-light photocatalytic performance and mechanism of zinc peroxide for tetracycline degradation and Escherichia coli inactivation, J. Colloid Interface Sci., 624 (2022) 137-149.

[7] W.G. Xu, J. Xu, Q.Y. Zhang, Z.P. Yun, Q.S. Zuo, L.P. Wang, Study on visible light photocatalytic performance of MIL-100(Fe) modified by carbon nanodots, Environ. Sci. Pollut. Res., 29  55069–55080.

[8] D.Y. Chen, Q. Bai, T.T. Ma, X.F. Jing, Y.Y. Tian, R. Zhao, G.S. Zhu, Stable metal-organic framework fixing within zeolite beads for effectively static and continuous flow degradation of tetracycline by peroxymonosulfate activation, Chem. Eng. J., 435 (2022) 134916.

Reviewer 2 Report

The aim of the paper “Natural diatomite supported zirconium doped TiO2 with tailoring band structure for enhanced visible-light photocatalytic properties” by Fang Yuan, Chunquan Li, Xiangwei Zhang, Renfeng Yang, and Zhiming Sun is to investigate the photocatalytic activity of composites based on Zr-doped TiO2 nanoparticles and diatomite. Similar research was reported for the degradation of other organic pollutants.

The manuscript is written in a clear manner and is well-structured and the experimental design is appropriate to test the hypothesis. The cited references are relevant to the field and the figures, tables, and images are appropriately illustrated and interpreted. However, the introduction section has to be improved by highlighting the novelty of the paper and the physicochemical analysis (XRD, SEM-EDS) could be better addressed. I recommend refining the manuscript based on the specific suggestions below:

Lines 87-88: Why was AgNO3 used for the synthesis? Where was it involved?

Lines 139-140: The unit of the diameter is wrong - “Figure 1a, the natural diatomite used in this work is in the shape of a disk with a diameter of about 20 m

Lines 147-152: SEM-EDS analysis: The authors state that all the elements are uniformly distributed in the diatomite disk area, but the analysis shows also that titanium is found in a higher concentration in the features on the left side of the image and on the right upper part of it. This means that there may be also titanium dioxide formed as individual particles. This effect may be caused by the synthesis procedure: the addition of solution B in solution A before the introduction of diatomite allows hydrolysis of the alkoxides before adsorption/absorption of the precursors on diatomite.

Lines 152-160: The interpretation of XRD analysis is poor. Evidence of crystalline doped Zr-TiO2 should be in the modification of the unit cell parameters as compared to pristine TiO2. Moreover, a change in the peak positions would be visible, most probably to lower angles as a consequence of doping TiO2 with a larger ion (Zr4+). Please consider evaluating the peak positions and/or the unit cell parameters.

Author Response

Response to Reviewer #2

The aim of the paper “Natural diatomite supported zirconium doped TiO2 with tailoring band structure for enhanced visible-light photocatalytic properties” by Fang Yuan, Chunquan Li, Xiangwei Zhang, Renfeng Yang, and Zhiming Sun is to investigate the photocatalytic activity of composites based on Zr-doped TiO2 nanoparticles and diatomite. Similar research was reported for the degradation of other organic pollutants.

The manuscript is written in a clear manner and is well-structured and the experimental design is appropriate to test the hypothesis. The cited references are relevant to the field and the figures, tables, and images are appropriately illustrated and interpreted. However, the introduction section has to be improved by highlighting the novelty of the paper and the physicochemical analysis (XRD, SEM-EDS) could be better addressed. I recommend refining the manuscript based on the specific suggestions below:

Comment 1:

Lines 87-88: Why was AgNO3 used for the synthesis? Where was it involved?

Response: Thanks for the reviewer’s comment. The AgNO3 is used to conduct the radical scavenger experiment to reveal the main radicals that played the major role within the photocatalytic process. In the experiments, the AgNO3 was employed to scavenge for the photocatalytic electrons (e). The corresponding contents in the manuscript were in Fig. 3d and line 237~248. Please check the revised manuscript.

Comment 2:

Lines 139-140: The unit of the diameter is wrong - “Figure 1a, the natural diatomite used in this work is in the shape of a disk with a diameter of about 20 m”.

Response: Thanks for the reviewer’s valuable comment. This is an error in the typesetting process using LaTeX. We are sorry for this careless mistake and have corrected it to μm in the revised manuscript. Please check the revised manuscript. The revised content is as follow:

As shown in Figure 1a, the natural diatomite used in this work is in the shape of a disk with a diameter of about 20 μm, and penetrating pores with a diameter of about 500 nm are regularly distributed on the surface.

Comment 3:

Lines 147-152: SEM-EDS analysis: The authors state that all the elements are uniformly distributed in the diatomite disk area, but the analysis shows also that titanium is found in a higher concentration in the features on the left side of the image and on the right upper part of it. This means that there may be also titanium dioxide formed as individual particles. This effect may be caused by the synthesis procedure: the addition of solution B in solution A before the introduction of diatomite allows hydrolysis of the alkoxides before adsorption/absorption of the precursors on diatomite.

Response: Thanks for the reviewer’s valuable comment. After careful consideration, we realized that the description of SEM-EDS was inaccurate. According to the suggestions of the reviewer, we added the explanation for the impact of this part in the manuscript. Please check the revised manuscript. The revised content is as follow:

In particular, there also exists TiO2 formed as individual particles, which might be attributed to the synthesis procedure: the addition of solution B in solution A before the introduction of diatomite allows hydrolysis of the alkoxides before the precursors loading on diatomite.

Comment 4:

Lines 152-160: The interpretation of XRD analysis is poor. Evidence of crystalline doped Zr-TiO2 should be in the modification of the unit cell parameters as compared to pristine TiO2. Moreover, a change in the peak positions would be visible, most probably to lower angles as a consequence of doping TiO2 with a larger ion (Zr4+). Please consider evaluating the peak positions and/or the unit cell parameters.

Response: Thanks for the reviewer’s valuable comment. After receiving the comments from the reviewer, we conducted XRD test on the prepared composite material samples again. The results show that there is no obvious peak position shift and the appearance of new characteristic diffraction peaks compared with pure TiO2. Therefore, in order to reveal the effect of Zr ion doping on the lattice structure of TiO2, we carried out high-resolution transmission microscope (HRTEM) on the 3%ZrTD composite (Figure 1d). The results show that its lattice spacing is 0.350 nm, which is consistent with the (101) crystal plane of anatase phase. Consequently, the Zr ion doping will not affect the lattice structure of TiO2 in the composite samples prepared by our proposed preparation process. The reason might be that the doping amount is too small. In addition, we also found from the relevant reports (ref: Ind. Eng. Chem. Res. 2018, 57, 14044−14051) that the Zr/TiO2 composite with low doping amount will not affect the lattice structure of TiO2. The revised Figure 1 is as follow:

Figure 1. SEM images of (a) D, (b) 3%ZrTD composite, (c) SEM-EDS element mapping of Zr, Ti and Si for 3%ZrTD composite, and the upper left corner shows the corresponding images; (d) HRTEM images of 3%ZrTD composite; (e) XRD patterns of D, T, TD composite, ZrT composite and 3%ZrTD composite; XPS spectra of (f) Ti 2p, (g) Si 2p, (h) Zr 3d and (i) O 1s of 3%ZrTD composite.

Round 2

Reviewer 1 Report

The manuscript is revised based on the comments.

Reviewer 2 Report

The manuscript has greatly improved. I recommend publication in present form.